:ᴏ: PLOS | ONE

# Cryo-focused ion beam preparation of perovskite based solar cells for atom probe tomography

**Nicolás Alfonso Rivas**[1,2,3]*, **Aslihan Babayigit**[1,2], **Bert Conings**[1,2], **Torsten Schwarz**[3], **Andreas Sturm**[3], **Alba Garzón Manjón**[3], **Oana Cojocaru-Mirédin**[4], **Baptiste Gault**[3,5], **Frank Uwe Renner**[1]

**1** Institute for Materials Research (IMO), Hasselt University, Diepenbeek, Limburg, Belgium, **2** IMEC vzw. Division IMOMEC, Diepenbeek, Limburg, Belgium, **3** Department of Microstructure physics and alloy design, Max-Planck-Institut für Eisenforschung GmbH, Düsseldorf, North Rhein-Westfalia, Germany, **4** Institut of Physics, RWTH Aachen University, Aachen, North Rhein-Westfalia, Germany, **5** Department of Materials, Royal School of Mines, Imperial College, London, United Kingdom

* nicolas.rivas@uhasselt.be

**Data Availability Statement:** All relevant data are within the paper.

## Abstract

Focused-ion beam lift-out and annular milling is the most common method used for obtaining site specific specimens for atom probe tomography (APT) experiments and transmission electron microscopy. However, one of the main limitations of this technique comes from the structural damage as well as chemical degradation caused by the beam of high-energy ions. These aspects are especially critical in highly-sensitive specimens. In this regard, ion beam milling under cryogenic conditions has been an established technique for damage mitigation. Here, we implement a cryo-focused ion beam approach to prepare specimens for APT measurements from a quadruple cation perovskite-based solar cell device with 19.7% efficiency. As opposed to room temperature FIB milling we found that cryo-milling considerably improved APT results in terms of yield and composition measurement, i.e. halide loss, both related to less defects within the APT specimen. Based on our approach we discuss the prospects of reliable atom probe measurements of perovskite based solar cell materials. An insight into the field evaporation behavior of the organic-inorganic molecules that compose the perovskite material is also given with the aim of expanding the applicability of APT experiments towards nano-characterization of complex organo-metal materials.

## Introduction

The uniqueness of atom probe tomography (APT) lies on its ability to analyze physical nano-structures and correlate them to the corresponding chemical information with sub-nanometer resolutions [1, 2]. State-of-the-art specimen preparation techniques have made APT increasingly amenable to analyze the chemistry and structure of a wide range of advanced functional and structural materials ranging from multi-layer devices used in the photovoltaic industry to metallic glasses for diverse applications [1–6]. In particular, the evolution of laser-assisted APT has permitted the analysis of materials with semiconducting properties since specimen

**Funding:** T.S. acknowledges the funding by the German Research Foundation (DFG) (Contract GA 2450/1-1).

**Competing interests:** The authors have declared that no competing interests exist.

requirement for high electrical conductivity may be circumvented by the use of light pulses to trigger field evaporation [1, 2, 7]. In this case, a local increase in temperature at the tip of the needle-shaped specimen subsequent to absorption of the laser light from the pulse provides the necessary energy for surface atoms to desorb from the lattice and, under the influence of the intense electrostatic field, ionize and accelerate towards the detector. This increase in temperature decreases the effective evaporation field. Thus, the probability of sample fracture attributable to the electrostatic pressure that originates from stress induced by the electrostatic field at the specimen surface is considerably reduced [2, 8].

One of the main challenges associated with analyzing organic materials with APT is the preparation of suitable specimens. Inevitably a needle-shaped specimen of approximately 100 nm diameter at its apex must be obtained to promote field evaporation within the range of working voltages accessible to commercial atom probe setups, i.e. 2–14kV. In this regard, two of the most commonly employed methods for preparation of organic samples for APT specimens [7–10] are: (1) electrochemical polishing of a metal wire with the desired shape and dimensions used as a substrate for depositing a thin layer of the actual organic material of interest; (2) site-specific or bulk lift-out and shaping by means of a focused ion beam (FIB), usually in a dual beam configuration within a scanning electron microscope (FIB-SEM).

In the case of FIB-based specimen preparation, and in particular for sensitive materials, great care must be taken in order to minimize ion–milling-induced damage. The conventional acceleration voltages (16-30kV), currents ($\geq$2.5 nA) that are used to successfully shape specimens from metallic materials can lead to amorphization, chemical degradation and formation of structural defects within organic specimens [11–13]. Additionally, the temperature increase on the sample surface induced by ion milling may also adversely affect the microstructure and consequently the properties of different type of materials [13–15].

In general, FIB milling with high energy ions (normally Ga) without the application of necessary safeguards induces overall degradation of organic materials from their pristine state [11, 16–19]. For this reason, one of the approaches used to considerably mitigate damage during FIB preparation is milling under cryogenic conditions, normally below 150 K. It is well established that cryo-FIB preparation modifies the ion sputtering mechanism on the specimen by limiting sample heating during the sputtering process [12, 20]. Consequently, Ga implantation during milling is reduced which better preserves the structure and integrity of sensitive materials including extremely delicate organic or biological specimens [20–23].

Here, we demonstrate how cryo-FIB preparation can be used to diminish beam damage of organo-metal halide perovskite based solar cells (PSC) for APT measurements. In particular, PSCs with mixed organic/inorganic cations such as methylammonium (MA, $CH_6N^+$), formamidinium (FA, $N_2H_5^+$), rubidium ($Rb^+$) and cesium ($Cs^+$) as well as mixed halide anions iodine ($I^-$) and bromine ($Br^-$) are studied. These materials have demonstrated remarkable performance, holding record power conversion efficiencies of almost 20% at the time of writing on par with established PV technologies while showing vast commercial potential in the photovoltaic landscape [24]. Nevertheless, some key challenges such as their toxicity and long term stability to the circumambient atmosphere remain to be resolved before their large-scale deployment. Hence, the intent of this work is to facilitate the usage of APT for analyzing photovoltaic perovskites by increasing the applicability horizon of the technique towards nano-characterization of sensitive organic-inorganic material systems.

## Materials and methods

Fabrication of PSC Solar Cells [25, 26]: Indium tin oxide (ITO) (On soda lime glass: Kintec) glass was patterned with Zn powder and 2 M HCl solution, and subsequently ultrasonically

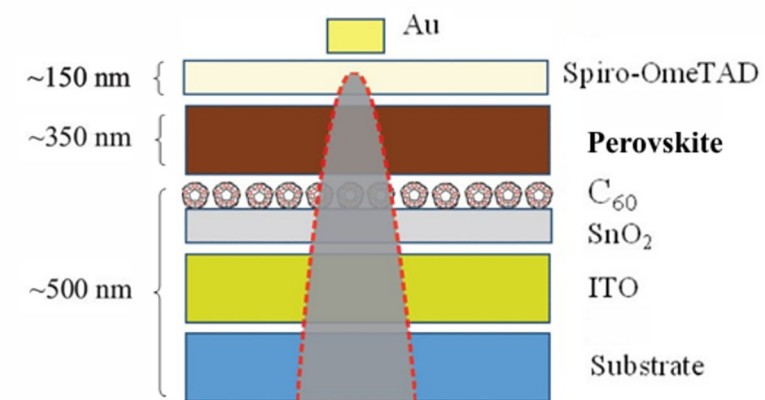

**Fig 1. Solar cell stack configuration and illustration of a final atom probe specimen.**

cleaned with Hellmanex soap solution, water, acetone, and 2-propanol (IPA). The samples were consequently treated with $O_2$ plasma for 5 min just before deposition of a 2% colloidal dispersion of SnO2 (Alfa Aesar: 44592) in $H_2O$. Afterwards, $C_{60}$ (Sigma Aldrich: 379646) was spin coated on the $SnO_2$ at 3500 rpm. The samples were then transferred to a nitrogen-filled glovebox. The 1.35 M perovskite precursor in DMF:DMSO (1:10) was spin coated and quenched by a strong $N_2$ flow directed towards the sample, as described elsewhere [25]. The source materials used for perovskite layer synthesis were: MAI (Greatcell Solar: MS101000), FAI (Greatcell Solar: MS150000), RbI (Alfa Aesar: 13497), $PbI_2$ (TCI: L0279), $PbBr_2$ (TCI: L0346) and CsI (Alfa Aesar:10992)

The samples were then heated for 60 min at 100˚C. Spiro-OMeTAD (Lumtec: LT-S922) was spin coated on top at 3000 rpm, after which the samples were stored in a desiccator overnight. Finally, an Au top contact ($\approx$100 nm) was deposited under high vacuum ($\approx 6 \times 10$–6 Torr). The resulting perovskite composition was $Rb_{0.05}$ [$Cs_{0.05}FA)_{0.83}$ ($MA)_{0.17}$) $_{0.95}]_{0.95}$ Pb ($I_{0.90}Br_{0.1})_3$ and the final solar cell stack architecture is illustrated in Fig 1.

In addition, these devices have remarkably exhibited excellent performance in terms of efficiencies of up to 19.7%±0.2% and a stabilized power output of 20% in their pristine state as shown in Fig 2 [25].

Preliminary imaging: Cross-sectional scanning electron microscopy (SEM) images from the photovoltaic device were obtained in immersion mode at 5 kV and 86 pA with a dual-beam FIB (FEI Helios Nanolab 600 and 600i).

APT needle-shaped specimen preparation: Specimens for APT were obtained with the use of a dual-beam FIB (FEI Helios Nanolab 600) system, mostly following the procedure described in Ref. [27]. In this case, however, for the final annular milling shaping steps a cryo-stage for liquid nitrogen cooling (Gatan C1001) inside the vacuum chamber was used to decrease the temperature of the specimen down to approximately -190 ᵒC. Moreover, since the cryo-stage cannot be tilted without damaging the SEM column pole piece due to the height of the cryo-stage, a holder was manufactured in-house as shown in (Fig 3A and 3B). This provided the necessary orientation for the carrier of the specimen with respect to the FIB column at 52ᵒ from the horizontal without the need for tilting the cryo-stage. Lift-outs were carried out randomly throughout the surface of the sample and through the entire depth of the solar cell stack as shown in Fig 1. FIB milling was performed using an accelerating voltage of 16kV and variable currents ranging from 0.13nA to 45 pA to further minimize Ga damage. A final milling step at a lower accelerating voltage of 5kV was performed in order to remove the

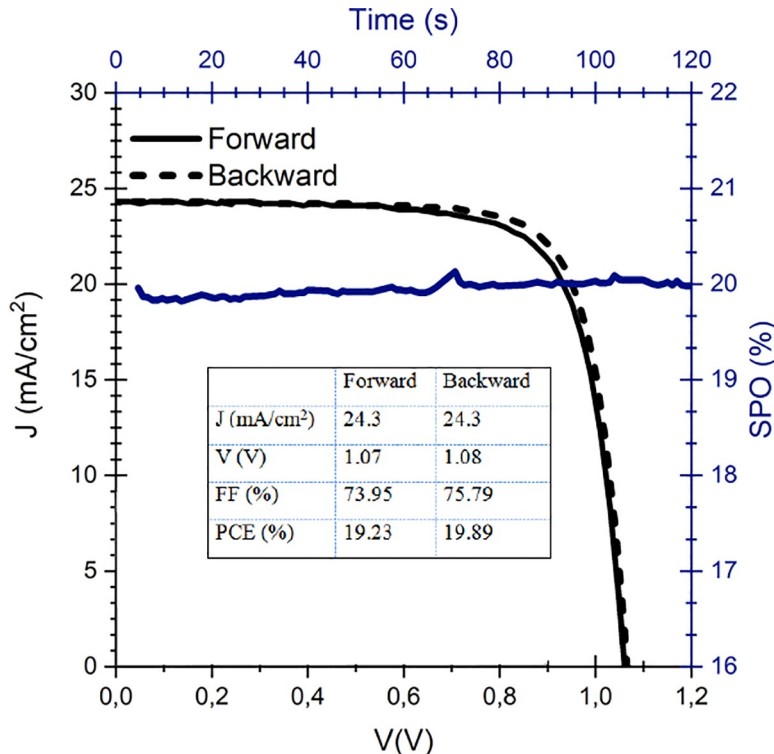

**Fig 2. J-V curve of the analyzed quadruple cation perovskite.** Inset: corresponding photovoltaic parameters.

regions of the specimen that had been severely damaged by the incoming energetic Ga ions, as documented in Ref. [27].

APT measurements were performed using a reflectron-fitted local electrode atom probe LEAP™ 5000XR (Cameca Instruments Inc.). The specimen base temperature was maintained at 50–60 K, and was analyzed in laser pulsing mode, with a UV-laser (355 nm wavelength), with laser pulse energies ranging from 1 pJ to 10 pJ. The laser repetition rate was kept at 65 kHz with an average detection rate of 5 ions per 1000 pulses.

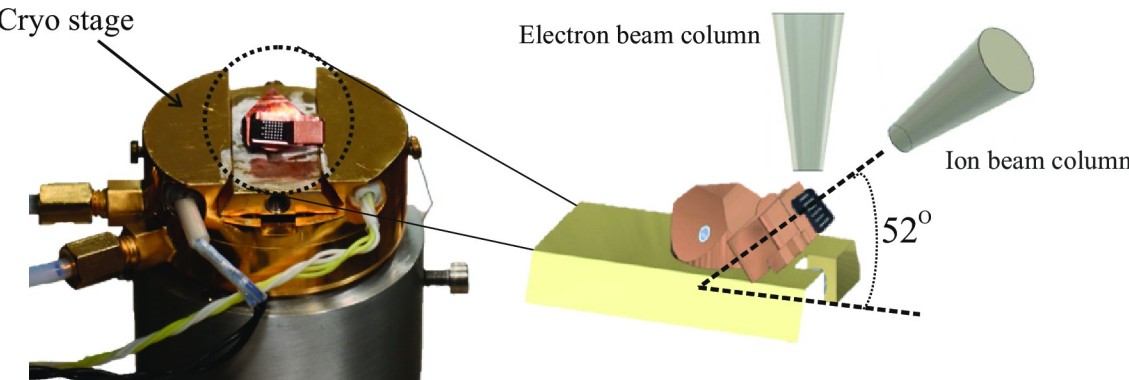

**Fig 3. In-house manufactured holder for cryo-stage annular milling of atom probe specimens.** (A) Holder mounted on the cryo-stage. (B) Schematic view of the holder within the dual-beam FIB.

## Results

### Preliminary sample characterization and specimen preparation

A cross section of the studied PSC stack is shown in Fig 4. Two notable characteristics of the films are immediately evident, as previously described in Ref. [26], and are important for the subsequent results obtained from the APT reconstructions: First, the perovskite layer is comprised of a polycrystalline structure with columnar grains of various sizes ranging from 50 nm to 400 nm (Fig 4). This makes characterizing a grain boundary with APT challenging since the FIB lift-outs are randomly obtained and the diameter of the final needle-shaped specimens are considerably smaller than the grain sizes. Second, the fullerene layer is observed as discreet islands, as seen in Fig 4, instead of an expected continuous film presumably due to the redissolution of $C_{60}$ upon perovskite deposition. Due to this film discontinuity, obtaining APT specimens that contain fullerenes was also difficult. Moreover, $C_{60}$ fullerenes decompose into fragments when exposed to the electric field required for evaporation [7]. In this case, if the laser energy is increased, $C_{60}$ may appear in the mass spectrum as ion fragments comprised of $C_{10}^{+}$ and $C_{18}^{+}$.

The sequence of steps that were followed for APT specimen preparation by means of FIB milling is shown in Fig 5. After a lift-out and successive welding of 2 μm-wide wedges onto Si microposts (Fig 5A and 5B), all the layers of the PSC remain clearly visible (Fig 5C). Nonetheless, as annular milling progresses and the tip of the specimen narrows, the layers cannot be resolved anymore as usually observed during milling of thin-film materials. Therefore, it is essential to monitor the annular milling process with a marker provided by the FEI-Helios software as shown in Fig 5D; in this case a line marker was used to stop annular milling when the Spiro-OmeTAD-Perovskite interface was positioned at the apex of the specimen (Fig 5E).

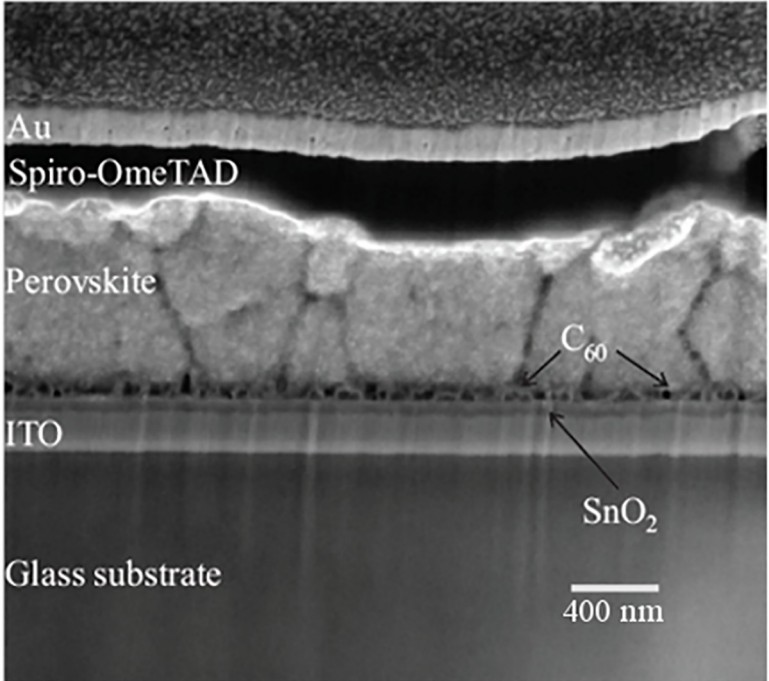

**Fig 4. SEM cross section image of the full PSC stack obtained by FIB.** Perovskite grains of several hundred nanometers and fullerene islands are observed.

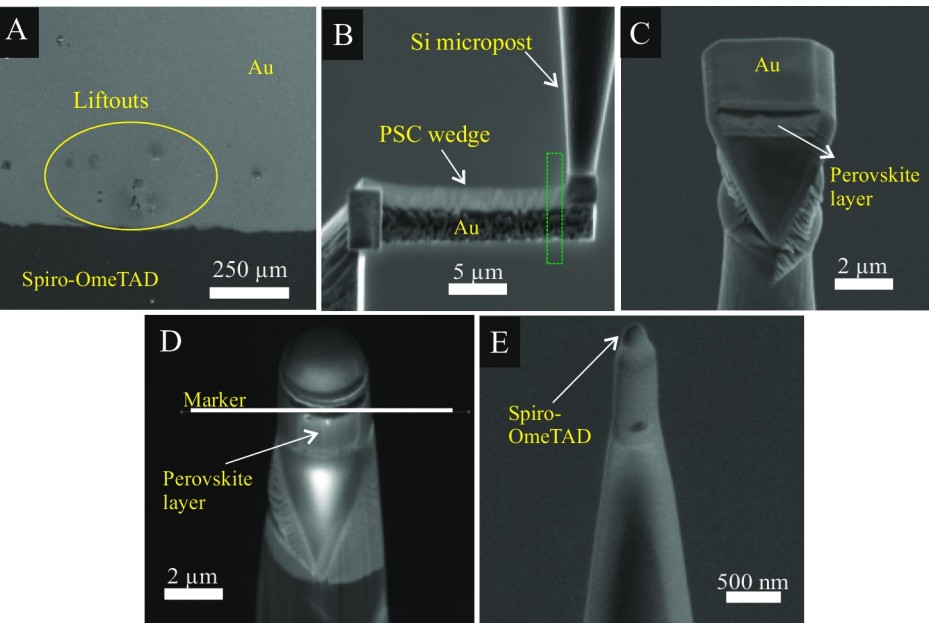

**Fig 5. APT specimen preparation of PSC.** (A) Wedges are cut from random spots across the surface of the Au contact and lifted out by the use of a micromanipulator. (B) The wedge is then soldered with Pt on Si microposts and cut in depth along the green box to obtain (C) 2 μm sized wedges before annular milling, the PSC layers are still clearly visible. (D) Marker used to manually control the end of annular milling. (E) Final specimen.

## APT results: Mass spectrum analysis and yield

Representative mass spectra from APT measurements from non-cryo and cryo-preparation specimens are shown in Fig 6A and 6B respectively. Some differences are readily observed: the specimen prepared at room temperature exhibits large amounts of unidentifiable peaks (Fig 6A). These peaks might originate from $C_xH_y$ fragments and/or hydrides ($BrH^+$,$PbIH^+$, $PbH^+$, etc.) induced by FIB milling at room temperature [7, 17].

On the contrary, the mass spectra from the cryo-milled samples (Fig 6B) exhibits easily identifiable peaks and considerably less background noise at high mass-to-charge state ratios. In this case, fragmentation of C-H chains as well as the formation of different hydrides seems to have been considerably reduced. It is possible that the Ga ions from FIB preparation at room temperature may have significantly modified the original perovskite structure and lead to a modified material with a different field evaporation behavior as compared to the cryo-milled samples.

Moreover, the $Si^+$ peak at 28 Da shown in Fig 6B originates from the glass substrate. This means that the APT experiment was long enough to reach the substrate, which is 1 μm below the surface of the device. In fact, the yield is significantly affected by preparing samples at room temperature. In this case, less than 40% of the experiments led to datasets with more than a 1 million detected ions, which were considered as successful, while for cryo-prepared samples more than 90% of the specimens were successfully analyzed. This may be due to defects introduced in the lattice by the high energy Ga ions from ion beam milling at room temperature. Since the perovskite material is highly susceptible to the electrostatic pressure generated during an APT experiment, any defect such as protrusions, holes or material discontinuity will severely cause stress concentrations in the material and lead to premature fracture of the specimen.

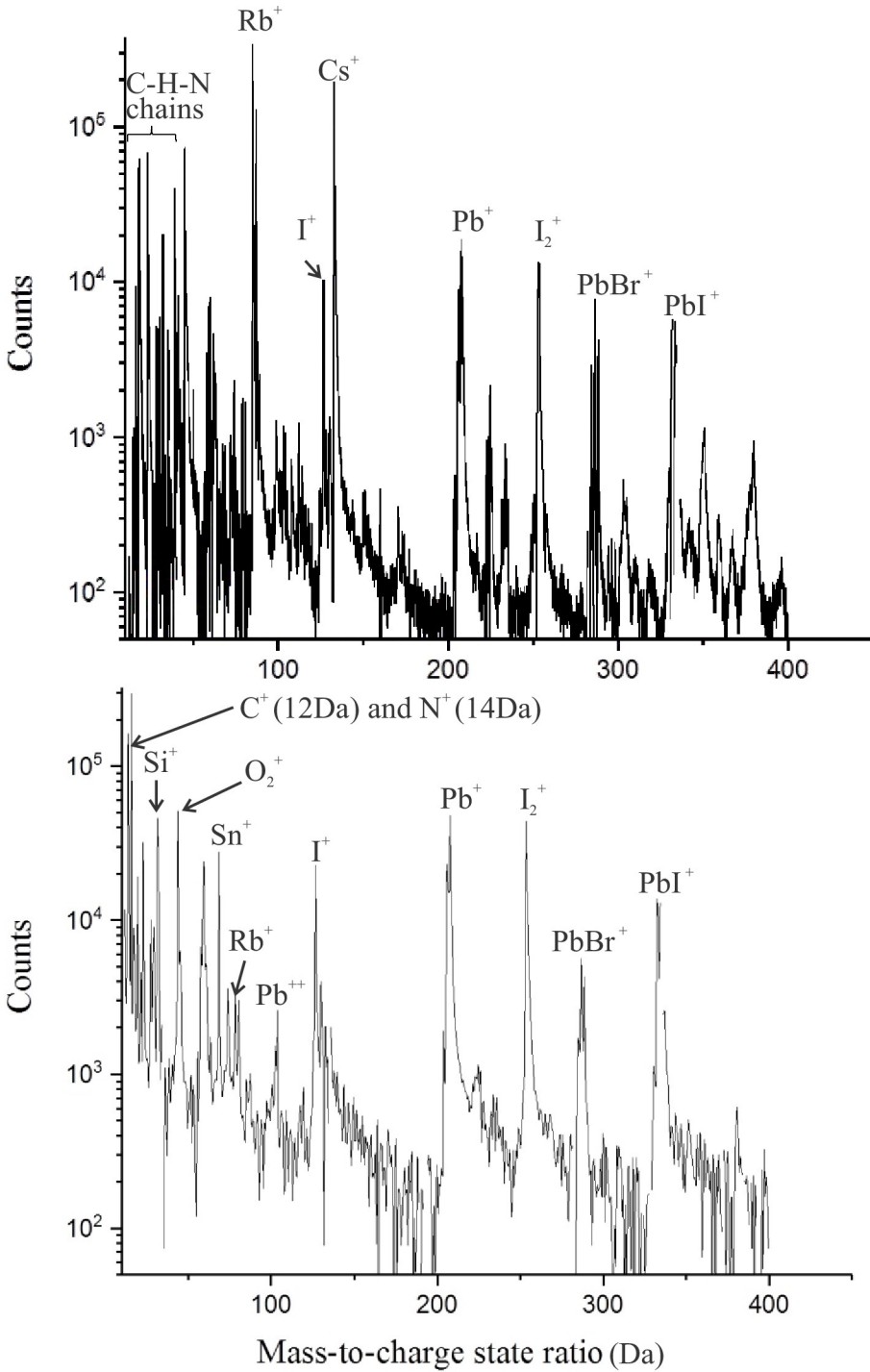

**Fig 6. Mass spectra obtained by APT (60 K, 1 pJ).** Main peaks are highlighted (A) Non-Cryo-prepared sample. (B) Cryo prepared sample.

In both conditions, the MA ($CH_6N^+$, 8 atoms) and FA ($CN_2H_5^+$, 8 atoms) ions are difficult to identify in the mass spectrum since in most cases these molecules are dissociated into fragments [7] such as $C_x$-$H_y$-$N_z$ compounds, $CH_2^+$, $CH_3^+$, $NH_3^+$, etc. between 6 and 50 Da. In

**Table 1. Summary of APT measurements obtained from non-cryo prepared samples.**

| Non-cryo prepared measurements | T (K) | Energy (pJ) | Result (ions) | Pb:I ratio |
|---|---|---|---|---|
| 1 | 55 | 10 | Fracture | - |
| 2 | 60 | 10 | 15.000.000 | 3:1 |
| 3 | 60 | 5 | Fracture | - |
| 4 | 60 | 10 | 6.200.000 | 1:1 |
| 5 | 50 | 10 | Fracture | - |
| 6 | 60 | 1 | 2.500.000 | 1,4:1 |
| 7 | 55 | 10 | Fracture | - |
| 8 | 55 | 15 | 5.100.000 | 2,5:1 |
| 9 | 60 | 10 | Fracture | - |

general, complex molecules can be formed under the high electrical fields generated in an APT experiment, particularly for organic hydrogen containing chains such as MA and FA. For this reason, providing spatial information from the organic chains that compose the perovskite will always be considerably challenging and beyond the scope of the present article. Nonetheless, the C and N counts determined by the peaks at 12 and 14 Da respectively are considerably higher in the cryo-prepared specimens as seen in Fig 6B. This suggests that the organic portion of the perovskite material was better preserved in specimens that were prepared under cryogenic conditions.

Finally, the laser pulsing frequency of the APT experiment must be carefully selected. Since the perovskite material is partially comprised of heavy molecular ions such as Rb, I, Cs and Pb, frequencies larger than 65 kHz results in these ions not being detected and forming part of the background noise due to uncorrelated detection events. This can also be observed in the mass spectrum throughout the experiment as a "wraparound" effect at low mass-to-charge state ratios. In addition, higher laser pulsing frequencies will prevent heat transfer from the surface of the specimen resulting in an increased base temperature, which consequently compromises the mass resolution of the measurement.

## Halide detection and distribution

The major problem with specimen preparation of a PSC by means of FIB is usually related to the loss of organic cations and halides [11, 28]. In particular, the measured iodine concentration after FIB milling at room temperature is considerably lower (5 at.% lower) than that expected from the stoichiometry of the perovskite film. Tables 1 and 2 show a summary of the ratio of the number of detected I ions to Pb ions obtained within different APT measurements performed on specimens prepared under non-cryo and cryo-conditions respectively. A clear trend was found: the nominal Pb:I ratio from 6 cryo-prepared specimens is mostly maintained

**Table 2. Summary of APT measurements obtained from cryo prepared samples.** The expected Pb:I ratio (1:2,7) is considerably improved when measuring samples prepared under this condition.

| Cryo-prepared measurements | T (K) | Energy (pJ) | Result (ions) | Pb:I ratio |
|---|---|---|---|---|
| 10 | 60 | 1–2 | 22.000.000 | 1:2,5 |
| 11 | 60 | 1 | 3.000.000 | 1:2 |
| 12 | 60 | 1 | 2.000.000 | 1:2 |
| 13 | 60 | 1 | 8.000.000 | 1:2,7 |
| 14 | 60 | 1 | 7.000.000 | 1:2,5 |
| 15 | 60 | 1 | 2.000.000 | 1:2,3 |

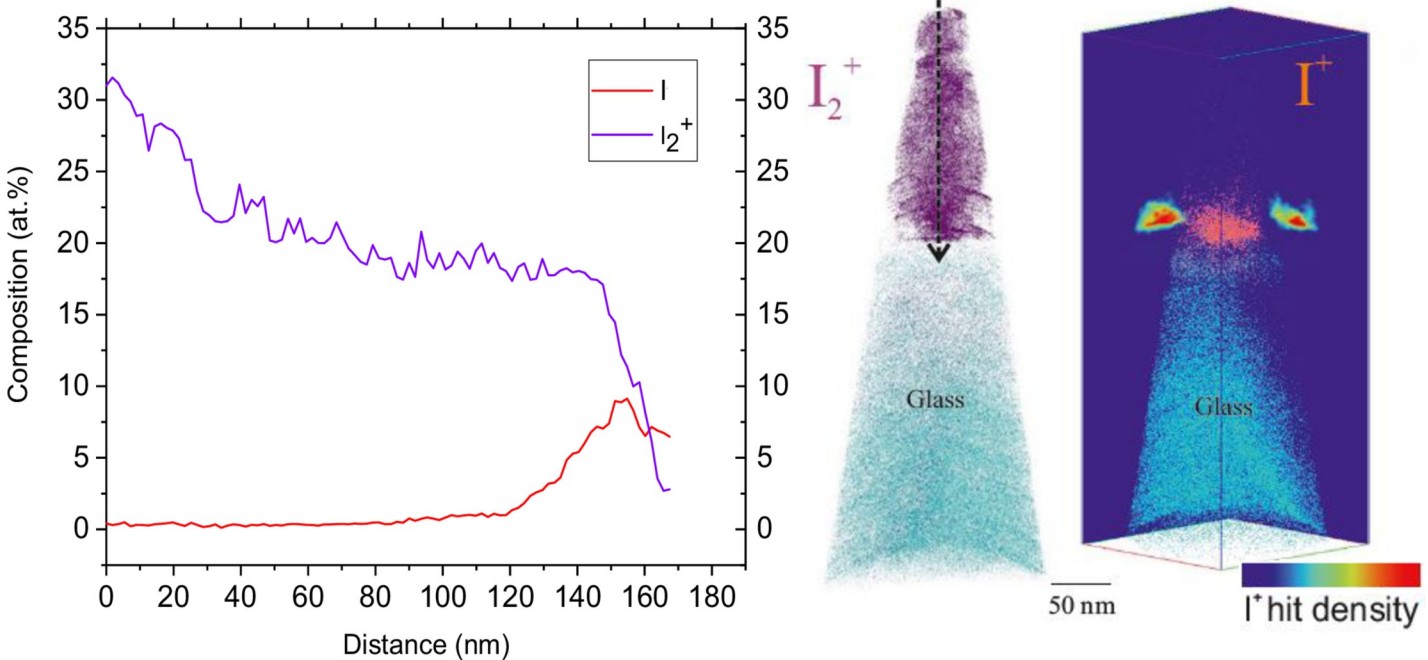

**Fig 7. Distribution of iodine within the perovskite film of a cryo-prepared specimen.** (A) Contribution of $I^+$ and $I_2^+$ on the total iodine concentration along the direction shown by the dashed arrow in (B). (B) Distribution of $I_2$ ions. (C) Distribution of I ions with the corresponding 2-D density plot on the X and Y planes.

(1:2.7±0.62) whereas specimens milled at room temperature exhibited significant loss of iodine, down to a ratio of 3:1. Additionally, since non-cryo prepared specimens were suscepti-ble to early fracture, higher laser energies were needed to reduce the electrostatic field required for evaporation. This likely had a detrimental effect on the measured composition of the perovskite due to excessive heating of the perovskite film.

We can conclude from these observations that the advantages of preparing APT specimens under cryogenic conditions are mainly twofold. First, due to the resistance of the specimens against early fracture under a high electrostatic field, as observed by yield, lower laser energies can be used throughout the experiment. This is mandatory since low laser energies are gener-ally required for analysis of photovoltaic materials characterized by high absorption capabili-ties as well as organic samples [29, 30]. Second, halide loss during FIB preparation is considerably mitigated. It is important to highlight that the main I contribution originates from the detection of the $I_2^+$ peak. $I_2$ may be present in the material in the form of $PbI_2$ as an ambient degradation product or from an excess added to the precursor during film processing [26, 31]. For this reason it is difficult to discern whether this molecule is present in the original sample or is formed as a complex ion during field evaporation. This may occur if the energy required for field evaporation of the $I_2$ molecule is less than that of I. However, Fig 7A shows the contribution of $I_2$ and I on the total iodine concentration of a representative cryo-prepared APT specimen. It shows a smooth decrease of I as the measurement progresses towards the ETL with a deviation from the nominal composition (24.2 at.%) of ±5at.%. This suggests that the iodine detected during field evaporation is originated from a homogeneous distribution of I within the perovskite film regardless of the nature of the parent iodine ion ($I_2$ or I). This result can be used for characterization of perovskite films that may contain inhomogeneities or phases with a surplus or depletion of iodine of more than 5 at.%.

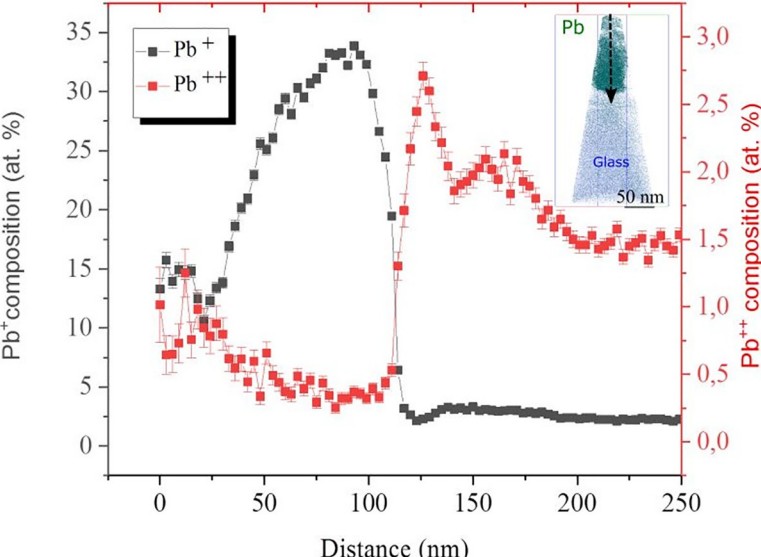

**Fig 8. Concentration of Pb$^+$ and Pb$^{++}$ along the direction indicated by the dashed arrow on the atom probe reconstruction (inset).** As field evaporation takes place towards the SnO$_2$-ITO layers a higher electric field is required to maintain the selected detection rate therefore leading to the post-ionization of Pb$^+$ into Pb$^{++}$.

In addition, as shown in Fig 7B and 7C, I$^+$ is consistently found towards the bottom part of the perovskite layer near the electron transporting layer (ETL). A few mechanisms may be contributing to this observed iodine behavior before and/or after the APT measurements. Firstly, the loss or degradation of I to I$_2$ may occur since the wedge is exposed to the high-energy Ga-ions or electron beam at room temperature during the early stages of FIB preparation [11, 31]. Most of this loss should, however, be inhibited by completing annular milling on the final atom probe specimen under cryogenic conditions. Secondly, iodine migration might occur within the perovskite film due to the high voltages used during an APT experiment. It has been shown that a bias of as little as 1V/μm applied to the perovskite material at temperatures above 0°C may induce drift of I towards its respective sink depending on device architecture [32]. Nonetheless, the APT experiment is performed at 60 K and, thus, the diffusion of I within the perovskite layer should be expected to be significantly impeded. As shown by Li et al. [33], only a considerable increase in the localized temperature due to laser pulsing during the measurement may be sufficient to provide the necessary driving force for I mobility under an electric field at extremely low temperatures. Lastly, this behavior may be explained by post dissociation of I$_2$ during the APT measurement.

The dissociation energy of a I$_2$ molecule is 1.68 eV [34]. This does not reflect the energy necessary to further dissociate a I$_2^+$ ion under intense electric field conditions, but can likely be used as a proxy. Since the radius of the specimen is larger towards the substrate and the temperature rise from the pulsed laser dissipates within a relatively larger volume, higher voltages are required to maintain the desired ion detection rate. Furthermore, higher electric fields are necessary in order to evaporate ions from the ETL and the glass substrate as the measurement proceeds towards the interface. The electric field at this point may then be high enough to increase the probability of I$_2$ dissociation. An approach that may be used for observing a change in electric field is by tracking charge state ratios for a specific element. Particularly, during field evaporation of the perovskite layer peaks at 102 Da., 103 Da, 103.5 Da. and 104 Da. appeared in the mass spectrum with natural abundance isotopic ratios corresponding to Pb$^{++}$.

Indeed, as can be observed in Fig 8 an increase in electric field is readily noticeable, namely, the Pb$^+$/Pb$^{++}$ ratio decreases as field evaporation continues towards the ETL. A general

increase of $Pb^+$ is observed across the length of the APT specimen due to inhomogeneity of Pb in the APT sample, nonetheless, the $Pb^{++}$ concentration increases by an order of magnitude as the APT experiment progresses towards the backside of the solar cell.

According to the post-ionization theory stated by Kingham [35] double ionization of Pb occurs at higher fields, therefore the sharp increase of the $Pb^{++}$ concentration directly correlates to a considerable increase in the electric field necessary to maintain the desired detection rate. In fact, Pb requires a 2nd ionization energy of 15.03eV which is more than double of that required for 1st ionization [36]. To further support this assertion, the second ionization potential for $I_2$ is 15.7 eV [37] similar to that of Pb. Therefore the appearance of $Pb^{++}$ towards the ETL layer signifies that it is highly likely that $I_2$ is also being ionized to form $I_2^{++}$, which under the effect of the electric field can be subsequently dissociated to form $I^+$ ions due to the relatively low binding energy of $I_2$.

Another consideration that must be noted as a consequence of these dissociative processes is the possible formation of neutrals. Depending on the laser energy, and therefore, the strength of the electric field on the surface of the specimen, dissociative events may occur near or away from the apex of the APT specimen. At higher laser energies, dissociation is most likely to occur far from the specimen surface where the probability of subsequent ionization of a neutral molecule is low due to the decreased intensity of the electric field [38]. These neutrals would consequently not be detected in an APT equipped with a reflectron system such as the LEAP™ 5000XR used in this work [39, 40]. This could account for the further loss of I as a neutral during the APT experiment at higher laser pulse energies, here observed with 10 pJ but also previously reported with energies between 30–100 pJ [28]. Furthermore, since the probability for ion dissociation within the perovskite layer is high as previously explained, neutrals may also form as a by-product of the successive dissociation of a parent ion. This behavior was previously observed during field evaporation of GaN where a particular dissociation pathway dissociated $GaN_3^{++}$ molecules and formed $N_2$ neutrals [40].

## Rb distribution

After having shown that the perovskite material is better preserved during cryo-FIB preparation it is possible to more reliably characterize its microstructure with APT measurements. In this regard, an important question for solar cell applications is the distribution of Rb within the perovskite film. In most of the specimens Rb was not detected with APT, nonetheless, in those specimens that did contain Rb it was seen to segregate in highly concentrated regions, both in the data from non-cryo and cryo-prepared specimens (not shown here). This experience suggests that Rb segregates in the perovskite film after synthesis. Segregation of Rb is moreover in line with a report in Ref [41] where Rb was suggested by NMR analysis not to be incorporated into the perovskite lattice but instead forms separate $RbI_xBr_{1-x}$ and $Rb_xPb_yBr_z$ phases. For solar cell applications it is yet unclear how these Rb-rich phases affect the efficiency of the photovoltaic device in the long term. A detailed study of these phenomena lies beyond the scope of this work. Nonetheless, further research is ongoing in order to investigate the evolution of the perovskite microstructure during different stages of its operating lifetime by means of APT and in correlation with other microscopic techniques and solar cell characterization. Organic-inorganic specimens that are susceptible to undesired changes due to electric fields, UV-light exposure and ultra-high vacuum may still lead to significant reconstruction artifacts due to the complex nature of the field evaporation process around APT specimens of these materials.

## Conclusions

Cryo-FIB was used to prepare samples of a quadruple-cation perovskite-based solar cell for investigation by APT. It was found that halide loss, in particular that of I, was mitigated due to

the diminished infiltration and damage of the perovskite film from high-energy Ga ions. However, further loss of I may occur during the APT experiment since formation of undetectable neutrals I or $I_2$ is possible due to dissociative events and the high sensitivity of the perovskite material towards laser energy.

In addition, the Iodide contribution in the mass spectrum mainly originated from detection of $I_2^+$. Since the $I_2^+$ ion forms probably during the measurement it is unclear whether this molecule is originally present in the perovskite film. $I^+$ directly was only detected when the APT tip was exposed to relatively large voltages, most likely due to dissociation effects caused by the increase of the electric field the tip is exposed to towards the $SnO_2$-ITO interface.

Regardless of the nature of the evaporated iodine ion, the total composition of iodine is mostly maintained with respect to the nominal values. This reliability can thus be used during APT characterization of perovskite films which may contain elemental segregation or phases with considerable deviations from the nominal composition.

Finally, milling at low temperatures has prevented the formation of defects in the material. The prepared APT specimens are more resistant to the elevated stresses caused by the electric field during an APT measurement and, by extension, provide a more-easily interpretable mass spectra. It can be concluded that cryo-FIB can be a viable solution to preserve the pristine nature of sensitive organic-inorganic materials for subsequent APT analysis. Nonetheless, care must be taken in interpreting the spatial distribution of elements in the reconstructions obtained from APT.

## Acknowledgments

The authors are grateful to Uwe Tezins and Volker Kree for their support to the APT, SEM, FIB, and TEM facilities at Max-Planck-Institut für Eisenforschung GmbH and Hans-Gerd Boyen at Hasselt University for discussions on perovskite solar cells. In part we used the LEAP 5000 HR at the Flemish APT center, KU Leuven.

## Author Contributions

**Conceptualization:** Oana Cojocaru-Mirédin, Baptiste Gault, Frank Uwe Renner.

**Data curation:** Nicolás Alfonso Rivas, Torsten Schwarz.

**Formal analysis:** Aslihan Babayigit, Bert Conings.

**Funding acquisition:** Frank Uwe Renner.

**Investigation:** Nicolás Alfonso Rivas, Torsten Schwarz, Alba Garzón Manjón.

**Resources:** Bert Conings, Andreas Sturm.

**Supervision:** Oana Cojocaru-Mirédin, Baptiste Gault, Frank Uwe Renner.

**Writing – original draft:** Nicolás Alfonso Rivas.

**Writing – review & editing:** Nicolás Alfonso Rivas, Aslihan Babayigit, Bert Conings, Torsten Schwarz, Alba Garzón Manjón, Oana Cojocaru-Mirédin, Baptiste Gault, Frank Uwe Renner.

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
