## [Decision Letter · Decision Letter 0]

24 Oct 2019

PONE-D-19-27172

Cryo-Focused Ion Beam Preparation of Perovskite based solar cells for Atom Probe Tomography

PLOS ONE

Dear Mr. Rivas,

Thank you for submitting your manuscript to PLOS ONE. After careful consideration, we feel that it has merit but does not fully meet PLOS ONE’s publication criteria as it currently stands. Therefore, we invite you to submit a revised version of the manuscript that addresses the points raised during the review process.

We would appreciate receiving your revised manuscript by Dec 08 2019 11:59PM. To enhance the reproducibility of your results, we recommend that if applicable you deposit your laboratory protocols in protocols.io, where a protocol can be assigned its own identifier (DOI) such that it can be cited independently in the future. For instructions see: http://journals.plos.org/plosone/s/submission-guidelines#loc-laboratory-protocols

We look forward to receiving your revised manuscript.

Kind regards,

Kalisadhan Mukherjee

Academic Editor

PLOS ONE

Journal Requirements:

1. Please ensure that your manuscript meets PLOS ONE's style requirements, including those for file naming. The PLOS ONE style templates can be found athttp://www.journals.plos.org/plosone/s/file?id=wjVg/PLOSOne_formatting_sample_main_body.pdf and http://www.journals.plos.org/plosone/s/file?id=ba62/PLOSOne_formatting_sample_title_authors_affiliations.pdf

2. Our internal editors have looked over your manuscript and determined that it is within the scope of our "Photovoltaic Solar Cell Materials - Design, Fabrication and Testing" Call for Papers. This collection of papers is headed by a team of Guest Editors for PLOS ONE: Maria Antonietta Loi (University of Groningen), Juan-Pablo Correa-Baena (Georgia Institute of Technology), Hongxia Wang (Queensland University of Technology), Shuxia Tao (Eindhoven University of Technology), Devid Fenning (University of California San Diego) and Graeme Blake (University of Groningen).

The Collection will encompass a diverse range of research articles on photovoltaic solar cells, including materials design, stability and testing.  Additional information can be found on our announcement page: https://collections.plos.org/s/solar-cells.

If you would like your manuscript to be considered for this collection, please let us know in your cover letter and we will ensure that your paper is treated as if you were responding to this call. If you would prefer to remove your manuscript from collection consideration, please specify this in the cover letter.

3. We recommend that you amend your methods section for reproducibility reasons so as to include the full details of the sources of all the materials used in you study (e.g suppliers, product numbers, etc.).

Additional Editor Comments (if provided):

Authors are suggested to comment on the repeatability/reproducibility of the performance.

If possible, authors are suggested to include relative standard deviations (RSD) of performance from 5-10 samples.

Reviewers' comments:

Reviewer's Responses to Questions

**Comments to the Author**

1. Is the manuscript technically sound, and do the data support the conclusions?

Reviewer #1: Yes

Reviewer #2: Yes

2. Has the statistical analysis been performed appropriately and rigorously? 

Reviewer #1: Yes

Reviewer #2: Yes

3. Have the authors made all data underlying the findings in their manuscript fully available?

Reviewer #1: Yes

Reviewer #2: Yes

4. Is the manuscript presented in an intelligible fashion and written in standard English?

Reviewer #1: Yes

Reviewer #2: Yes

5. Review Comments to the Author

Reviewer #1: In the manuscript entitled “Cryo-Focused Ion Beam Preparation of Perovskite based solar cells for Atom Probe Tomography” by Nicolás Alfonso Rivas et al. report a cryo-focused ion beam approach to prepare specimens for atom probe tomography measurements from a quadruple cation perovskite-based solar cell device with 19.7% efficiency. As opposed to room temperature FIB milling, authors found that cryo-milling considerably improved APT results in terms of yield and composition measurement, i.e. halide loss, both related to less defects within the APT specimen.

The data is quite thorough in this study. I think this work is of general interest to the reader of PLos one in the field of device fabrication of complex organo-metal materials.

Reviewer #2: The manuscript entitled” Cryo-Focused Ion Beam Preparation of Perovskite based solar cells for Atom Probe Tomography” works on using Cryo-FIB to prepare samples for hybrid perovskite solar cells. The organization of the data and language in this manuscript is well organized and clear expressed. The detailed operation process and the results obtained from the measurement are helpful for the understanding of perovskite materials. Thereby, I do recommend publishing this manuscript after minor revision.

1. Are there some reports on sample preparation of hybrid perovskite film? If yes, please make comparison to confirm the advantages with cryo.

2. The fullerene layer is observed as discreet islands as mentioned in the manuscript. Does the so-called island formed during the sample preparation using FIB? Please supply the cross-sectional images for samples prepared by simple break rather than FIB.

6. PLOS authors have the option to publish the peer review history of their article (what does this mean?). If published, this will include your full peer review and any attached files.

Reviewer #1: No

Reviewer #2: No

---

## [Author Response · Author response to Decision Letter 0]

8 Dec 2019

Dear reviewer #2:

Thank you for reviewing our manuscript with title “Cryo-focused ion beam preparation of perovskite based solar cells for atom probe tomography”. Below are the responses to your requested minor revisions and the raised questions: 

1. There are indeed to the best of our knowledge only two reports on the context of sample preparation of perovskite based solar cells and these mentioned only the initial MAPbI3. These two publications, [1] and [2] respectively, focus on preparation for TEM analysis and also include the use of (room-temperature) Focused Ion Beam (FIB) preparation which is in principle also the way to prepare APT specimens in manuscripts: For the specific case of atom probe specimen preparation, only the one report is thus currently available [2]. Yet, both reports do not include cryo-preparation, which is the main result that we present here.

Their main observations are consistent with our initial starting point, namely, a considerably halide loss is observed after analyzing perovskite material with APT specimens prepared under standard (temperature) conditions. We thus provide an very important improvement on these results by preparing atom probe specimens under cryogenic conditions and report a novel quality achieved which, although still not perfect as we describe but allowing these studies now even with quantitative aspects as we describe in our manuscript.

In [1] Rothmann et.al establish that CH3NH3PbI3 solar-cell-based materials undergo considerable damage when analyzed under electron beam (SEM) and milled with focused ion beam at standard operating temperatures. In general, they observe that the perovskite layer endures continuous structural and compositional changes. Specifically, the perovskite material loses iodine (I) down to a ratio Pb:I of 1:2 instead of that expected from the stoichiometry (1:3). They also comment on the effect of analyzing MAPbI3 with TEM under cryo conditions. They observe that perovskite undergoes amorphization but no observations were documented regarding the final halide concentration. They also mention that the effect of grain boundary changes during TEM investigation is not as drastic as when measuring at standard temperatures. Moreover, specific Ga-focused ion beam interaction with the perovskite sample at liquid nitrogen temperatures was not discussed. We here now propose, by measuring the final composition with atom probe tomography, that the volatilization of halide is reduced when samples are prepared under cryo-conditions. Finally, Rothmann et.al only analyzed MAPbI3 based perovskite layers, in our case multiple cations (Rb, Cs and FA) are added to the perovskite chemistry in order to stabilize the perovskite structure and provide longer term stability for solar cell applications [3, 4]. 

2. In the figure below, a non-FIB milled, simply-cleaved cross section is shown. Since this sample was set for EBIC measurements the magnification was kept low to avoid structural damage to the perovskite film. Nonetheless, the discreet island features of the C60 fullerene layer are still observable. We ask not to publish this image in this manuscript since this is confidential work of some of our collaborators in an independent own study with industry participation. In answer to the reviewer we therefore propose that the observed discreet islands are not formed during FIB milling but rather during the synthesis process, also since the solubility of C60 in the solvent is quite low [5] which could therefore indeed lead to a higher propensity of agglomeration upon the spin coating process. 

[1] M. U. Rothmann, W. Li, Y. Zhu, A. Liu, Z. L. Ku, U. Bach, et al., "Structural and Chemical Changes to CH3NH3PbI3 Induced by Electron and Gallium Ion Beams," Advanced Materials, vol. 30, Jun 20 2018.

[2] N. A. Kotulak, Z. S. Almutawah, S. C. Watthage, A. Phillips, D. W. Zhao, K. E. Knipling, et al., "3D imaging compositional map in one-step growth of CH3NH3PbI3," 2018 Ieee 7th World Conference on Photovoltaic Energy Conversion (Wcpec) (a Joint Conference of 45th Ieee Pvsc, 28th Pvsec & 34th Eu Pvsec), pp. 0485-0489, 2018.

[3] M. Saliba, T. Matsui, J. Y. Seo, K. Domanski, J. P. Correa-Baena, M. K. Nazeeruddin, et al., "Cesium-containing triple cation perovskite solar cells: improved stability, reproducibility and high efficiency," Energy & Environmental Science, vol. 9, pp. 1989-1997, 2016.

[4] O. A. Syzgantseva, M. Saliba, M. Gratzel, and U. Rothlisberger, "Stabilization of the Perovskite Phase of Formamidinium Lead Triiodide by Methylammonium, Cs, and/or Rb Doping," Journal of Physical Chemistry Letters, vol. 8, pp. 1191-1196, Mar 16 2017.

[5] C. I. Wang and C. C. Hua, "Solubility of C-60 and PCBM in Organic Solvents," Journal of Physical Chemistry B, vol. 119, pp. 14496-14504, Nov 12 2015.

---

## [Decision Letter · Decision Letter 1]

3 Jan 2020

Cryo-focused ion beam preparation of perovskite-based solar cells for atom arobe tomography

PONE-D-19-27172R1

Dear Dr. Rivas,

We are pleased to inform you that your manuscript has been judged scientifically suitable for publication and will be formally accepted for publication once it complies with all outstanding technical requirements.

With kind regards,

Kalisadhan Mukherjee

Academic Editor

PLOS ONE

Additional Editor Comments (optional):

Reviewer 1 and Reviewer 2 both have recommended the acceptance of the article. The manuscript thus may be accepted for publication.

Reviewers' comments:

Reviewer's Responses to Questions

**Comments to the Author**

1. If the authors have adequately addressed your comments raised in a previous round of review and you feel that this manuscript is now acceptable for publication, you may indicate that here to bypass the “Comments to the Author” section, enter your conflict of interest statement in the “Confidential to Editor” section, and submit your "Accept" recommendation.

Reviewer #2: All comments have been addressed

2. Is the manuscript technically sound, and do the data support the conclusions?

Reviewer #2: Yes

3. Has the statistical analysis been performed appropriately and rigorously? 

Reviewer #2: Yes

4. Have the authors made all data underlying the findings in their manuscript fully available?

Reviewer #2: Yes

5. Is the manuscript presented in an intelligible fashion and written in standard English?

Reviewer #2: Yes

6. Review Comments to the Author

Reviewer #2: (No Response)

7. PLOS authors have the option to publish the peer review history of their article (what does this mean?). If published, this will include your full peer review and any attached files.

Reviewer #2: No

---

## [Editor Report · Acceptance letter]

8 Jan 2020

PONE-D-19-27172R1 

Cryo-focused ion beam preparation of perovskite based solar cells for atom probe tomography 

Dear Dr. Rivas:

I am pleased to inform you that your manuscript has been deemed suitable for publication in PLOS ONE. Congratulations! Your manuscript is now with our production department. 

With kind regards,

on behalf of

Dr. Kalisadhan Mukherjee 

Academic Editor

PLOS ONE